# Unsupervised 3D Reconstruction with Multi-Measure and High-Resolution Loss

**DOI:** 10.3390/s23010136

**Published:** 2022-12-23

**Authors:** Yijie Zheng, Jianxin Luo, Weiwei Chen, Yanyan Zhang, Haixun Sun, Zhisong Pan

**Affiliations:** College of Command and Control Engineering, Army Engineering University of PLA, Nanjing 210007, China

**Keywords:** three-dimensional reconstruction, multi-view reconstruction, deep learning, PatchMatch, unsupervised learning, feature point consistency, high-resolution loss

## Abstract

Multi-view 3D reconstruction technology based on deep learning is developing rapidly. Unsupervised learning has become a research hotspot because it does not need ground truth labels. The current unsupervised method mainly uses 3DCNN to regularize the cost volume to regression image depth. This approach results in high memory requirements and long computing time. In this paper, we propose an end-to-end unsupervised multi-view 3D reconstruction network framework based on PatchMatch, Unsup_patchmatchnet. It dramatically reduces memory requirements and computing time. We propose a feature point consistency loss function. We incorporate various self-supervised signals such as photometric consistency loss and semantic consistency loss into the loss function. At the same time, we propose a high-resolution loss method. This improves the reconstruction of high-resolution images. The experiment proves that the memory usage of the network is reduced by 80% and the running time is reduced by more than 50% compared with the network using 3DCNN method. The overall error of reconstructed 3D point cloud is only 0.501 mm. It is superior to most current unsupervised multi-view 3D reconstruction networks. Then, we test on different data sets and verify that the network has good generalization.

## 1. Introduction

Three-dimensional reconstruction refers to the establishment of a mathematical model suitable for computer representation and processing of three-dimensional objects. It has been widely used in the fields of virtual reality, robotics, automatic driving, and land resource utilization [1,2]. Multi View Stereo (MVS) [3] restores dense representations of three-dimensional scenes from multi-view images and calibrated cameras. This has always been an important subject in 3D reconstruction. MVS has been extensively studied for decades.

There are many traditional methods, including voxel-based [4,5], feature point diffusion [6,7], and depth map fusion [8,9]. Voxel-based methods divide the 3D space into regular grids, and then estimate whether each voxel is attached to the surface. The disadvantages of this representation are spatial discretization errors and high memory consumption, and its accuracy mainly depends on the resolution of the voxel. In the method based on feature point diffusion, the 3D coordinates of feature points are first generated from the matched image. Then, the 3D points are expanded to the spatial neighborhood according to the principle of luminosity consistency and visibility. In this way, the blank area may be seriously affected by the texture-free problem. Depth map fusion decomposes the complex MVS problem into a relatively small per-view depth map estimation problem, focusing on one reference and several source image depth maps at a time, and then all depth maps are fused together to form the final point cloud.

The traditional methods have been mature and achieved good results in scene reconstruction from Lambertian surfaces. However, in the case of illumination variation, low texture area, and non-Lambertian surface, the reconstructed match is not reliable. Aanæs et al. [10] and Knapitsch et al. [11] evaluated some mainstream MVS algorithms and found that although the most advanced traditional methods performed very well in accuracy, there was still a lot of room for improvement in the completeness of reconstruction.

In recent years, with the rapid development of deep learning neural networks, a large number of researchers have tried to introduce deep learning into MVS. Earlier supervised learning MVS include SurfaceNet [12] and LSM [13]; however, SurfaceNet and LSM are limited to small-scale reconstruction due to common shortcomings of voxel representation. They either employ a divide-and-conquer strategy or are only applicable to synthetic data with low resolution inputs. Recent deep learning-based methods mainly use Convolutional Neural Network (CNN) to infer the depth map of each view and then carry out a separate multi-view fusion process to build a 3D model. The most representative one is MVSNet, an end-to-end deep learning architecture proposed by Yao et al. in 2018 [14]. The main idea is to build a cost volume based on the plane scanning process, and then conduct 3DCNN regularization to return the depth. It computes a depth map one at a time instead of the entire 3D scene. The accuracy and completeness of the network reconstruction point cloud and the generalization ability of the network are better than most 3D reconstruction methods of the same period. Therefore, this method is widely used for depth estimation in most deep learning MVS networks [15,16,17,18,19,20]. However, to ensure the accuracy of depth calculation, the storage requirement is three times that of image resolution. So, it is difficult to process high-resolution images, which is also a common problem of 3DCNN regularization. To solve this problem, Yao et al. [21] replaced 3DCNN with cyclic regularization based on GRU in 2019. They sequentially adjusted 2D cost maps along the depth direction by gated recursive units (GRUs). It greatly reduced memory consumption and made high-resolution reconstruction possible. However, this results in memory reduction and increased running time.

In 2019, Luo et al. [22] improved the MVSNet cost volume by using piecewise confidence aggregation to improve the matching accuracy and robustness. In 2020, Gu et al. [23] and Yang et al. [16] both used the coarse-to-fine method to construct the cost volume pyramid. The result is a compact, lightweight network. The network can predict high-resolution depth maps to obtain better reconstruction results. The network computing speed has been greatly improved. Different from the cost-volume method of MVSNet, Chen et al. [24] proposed a point-based deep network framework. It directly processed the target scene as a point cloud to predict depth in a coarse-to-fine way. Although this method has a good reconstruction effect, its run time increases almost linearly with the iteration level.

Different from previous methods, Wang et al. [25] introduced the idea of PatchMatch [26] into MVS network and proposed PatchmatchNet, a supervised network. In addition, a learnable and adaptive module was adopted to improve the transmission mode and cost calculation of PatchMatch. The results show that compared with the previous 3D cost volume using rules, the proposed method has faster speed and lower memory requirements. It can also process high-resolution images. So, it is more suitable for resource-limited devices.

Supervised learning MVS networks need to use the true depth of the image as a baseline and compare it with the depth predicted by the network to calculate the error. This approach requires accurate image depth information in the training set. This depth information needs to be either obtained by high-precision instruments such as laser scanners or calculated using traditional geometric methods. The former needs a lot of manpower and financial resources, the latter is easy to have a larger error. Because of this, it is difficult to produce high-quality MVS data sets. Therefore, there are not many MVS data sets available today. However, unsupervised learning can solve these problems well. It does not require image depth information. It constructs the loss function by the relation of luminosity, feature, and semantics among images from different perspectives. Therefore, unsupervised learning can be adapted to more data sets. It also reduces the difficulty of making MVS data sets.

Unsupervised learning MVS usually uses photometric consistency or geometric consistency as self-supervised signals. In 2019, Khot et al. [17] first proposed an unsupervised MVS framework, Unsup_MVS. It used a method like MVSNet to predict depth and used the photometric consistency among multiple views as the supervised signal. Most previous MVS infer depth maps only for reference images, while the MVS2 network proposed by Dai et al. [18] is symmetric for all views. It treats each view equivalently while predicting the depth map of each view. In 2020, Huang et al. [19] proposed an unsupervised multi-metric network framework M3VSNet. In this architecture, multi-scale pyramid feature aggregation is used to construct a 3D cost volume with more context information, and the loss function combines pixel loss and feature loss. In 2021, Xu et al. [20] combined data augmentation and semantic segmentation as self-supervised signals, making the reconstruction effect comparable to that of the most advanced supervised learning networks. Yang et al. [27] comprehensively used various methods such as deep fusion, mesh generation and deep rendering in unsupervised networks to optimize the pseudo depth.

Inspired by the above research, we propose an unsupervised multi-view 3D reconstruction network framework named Unsup_patchmatchnet based on PatchMatch. The network depth estimation module is the same as the main part of the supervised network PatchmatchNet [25], both of which adopt multi-scale depth prediction. The self-supervised method adopts the multi-metric union. It integrates the consistency of photometric, semantic, and feature points. Photometric consistency has always been considered the most basic supervision signal. It is assumed that the photometric of the same point or patch from different perspectives will hardly change. The current unsupervised MVS networks mainly adopt this method. Semantic consistency can provide abstract matching clues to guide supervision and can enhance robustness to color fluctuations. In this paper, the non-negative matrix factorization method used in reference [20] is adopted for semantic consistency. Therefore, this part does not need to be pre-trained and has good robustness to different scenes. Feature point detection and matching is a common method in traditional 3D reconstruction. The feature points generally have good robustness and can adapt to the influence of rotation, scale, and illumination. Feature point consistency utilizes the alignment of feature points matched between different perspectives to guide depth refinement. Compared with a large number of image pixels, there are fewer feature points between images. The image depth of the location of feature points can be estimated quickly by using the consistency of feature points. PatchMatch can quickly spread the correct depth to the surrounding points. Therefore, the network will be more efficient and accurate. In addition, depth maps with multiple resolutions are generated in the process of multi-scale depth estimation, and higher resolution images are input in model testing or practical applications. Therefore, we adopt the high-resolution loss method. We up-sample the depth maps estimated at each stage of the network to the maximum resolution, and then calculate the loss of various measures.

Our main contributions are summarized as follows:

(1) In this paper, we propose for the first time an end-to-end unsupervised multi-view 3D reconstruction network Unsup_patchmatchnet based on PatchMatch. The network incorporates multi-metric self-supervised signals.

(2) Based on the detection and matching of feature points in traditional methods, we propose the feature point consistency loss as the self-supervised signal of unsupervised MVS for the first time. 

(3) For the first time, we propose a method to use high-resolution loss instead of multi-scale loss. The ablation experiment in Section 3.2.1 verifies that it can further improve the network performance.

## 2. Materials and Method

The proposed Unsup_patchmatchnet network structure is shown in Figure 1. The network can be divided into two main parts. The first part is a multi-scale depth estimation based on PatchMatch. The second part is the multi-metric high-resolution joint loss which fuses the photometric consistency loss, semantic consistency loss, and feature point consistency loss.

### 2.1. Multi-Scale Depth Estimation Based on PatchMatch

The depth estimation module adopts the backbone part of Patchmatchnet [25]. This section has not been significantly changed. This module can be divided into four stages, estimating depth maps of four different scales, and gradually refining the depth from coarse to fine. The multi-scale features are extracted first, and then the depth is estimated at each scale. The process of each scale is basically the same. Starting from the smallest scale (stage 3), depth initialization is performed first, that is, sampling within the image depth range of [*d*_min_, *d*_max_]. Each subsequent iteration or upsampling to a new scale is not initialized, but a small random perturbation is added to the depth. Adaptive propagation is to add the current depth value of a pixel that has the same depth as that pixel to the depth assumption of that pixel. The adaptive evaluation is to calculate the weight of each depth hypothesis for each pixel and regress the depth of each pixel. The depth map generated at the lower scale is upsampled as the initial depth value at the higher scale. In stage 0, propagation is no longer carried out, but the multi-scale guided convolution network MSG-Net [28] is used to up-sample the depth map and generate the final depth map.

### 2.2. Multi-Measure and High-Resolution Loss

Unsup_patchmatchnet integrates a variety of measurement losses such as photometric consistency loss, semantic consistency loss, and feature point consistency loss.

#### 2.2.1. Photometric Consistency Loss

Photometric consistency [29] means that the same object should have the same color value from different perspectives. Photometric consistency is the most common method for calculating losses in unsupervised MVS networks.

As shown in Figure 2, assuming that pixel $p_{j}$ on the reference image corresponds to pixel $p_{j}^{\prime }$ in the source image, $p_{j}^{\prime }$ can be calculated by the following formula:(1)$$p_{j}^{\prime }=KT\left(D\left(p_{j}\right)K^{-1}p_{j}\right)$$
where *j* represents the pixel index, *K* is the internal reference matrix, *T* is the projection matrix from the reference perspective to the source perspective, and *D* is the depth.

Thus, image $I_{i}^{\prime }$ which is warped from the source image to the reference perspective can be constructed.
(2)$$I_{i}^{\prime }\left(p_{j}\right)=I_{i}(p_{j}^{\prime })$$
where *i* is the index of the source view.

The photometric consistency loss is the sum of the photometric loss of each reference image and all related source images.
(3)$$L_{PC}=\overset{N}{\underset{i=2}{\sum }}\frac{∥\left(I_{i}^{\prime }-I_{1}\right)⊙M_{i}{∥}_{2}+∥\left(\nabla I_{i}^{\prime }-\nabla I_{1}\right)⊙M_{i}{∥}_{2}}{∥M_{i}{∥}_{1}}$$
where ∇ represents the gradient, ⊙ is the dot product, *N* represents the number of source images, $M_{i}$ represents the binary validity mask of the *i*-th source image. The mask of the *j*-th pixel $p_{i,j}$ of the *i*-th source image is calculated as follows:
(4)$$M_{i,j}=\left\{\begin{array}{l} 1,p_{i,j}\in I_{i}^{\prime }\cap I_{1}\\ 0,else \end{array}\right.$$

#### 2.2.2. Semantic Consistency Loss

In reality, due to the different illumination conditions and the reflected light of the object, the same object in different viewing angles will have inconsistent luminosity. Figure 3 shows the images of a scene in the DTU dataset from three different perspectives under the same lighting condition. The same object has different luminosity at different angles. The difference is even more pronounced if the lighting conditions are different. Therefore, we introduce semantic consistency loss to compensate for the lack of photometric consistency loss function.

Semantic consistency means that the corresponding pixels of different images should have the same semantic type. In this paper, the method described in the literature [20] is adopted. Firstly, the pre-trained VGG16 network is used to extract image features, and then the non-negative matrix factorization method is used to cluster image pixels to generate a semantic segmentation map. As shown in Figure 4, the calculation method is similar to the photometric consistency loss. According to Equation (1), the pixel $p_{j}$ on the reference image is calculated and the corresponding pixel $p_{j}^{\prime }$ in the source image is calculated. Then the warped segmentation map $S_{i}^{\prime }$ from the *i*-th source view is reconstructed by bilinear sampling.
(5)$$S_{i}^{\prime }\left(p_{j}\right)=S_{i}(p_{j}^{\prime })$$

Finally, the cross-entropy loss of each pixel between the warped segmentation map $S_{i}^{\prime }$ and the reference segmentation map $S_{1}$ is calculated as the semantic consistency loss.
(6)$$L_{SC}=-\overset{N}{\underset{i=2}{\sum }}\left[\frac{1}{∥M_{i}{∥}_{1}}\overset{HW}{\underset{j=1}{\sum }}f\left(S_{1,j}\right)\log(S_{i,j}^{\prime })M_{i,j}\right]$$
where, $f\left(S_{1,j}\right)=\mathrm{onehot}\left(\mathrm{argmax}\left(S_{1,j}\right)\right)$, $M_{i}$ represents the binary validity mask of the *i*-th source image, the calculation is the same as in Section 3.2.1.

#### 2.2.3. Feature Point Consistency Loss

The boundary of pixel semantic clustering in the image is not accurate enough, while the position of image feature points is relatively accurate. The network can be guided to predict more accurate image depth by the position offset of the matched feature point pairs between the reference image and the source image.

The feature point consistency requires that the position of the feature point projected into the source image from the reference image should be the same as the position of the feature point matched in the source image. As shown in Figure 5, The feature point $p_{1}$ in the reference image matches the feature point $p_{i}$ in the source image. The position of $p_{1}$ projected into the source image through Equation (1) is $p_{i}^{\prime }$. According to the requirement of feature point consistency, $p_{i}$ and $p_{i}^{\prime }$ should be in the same position. However, in practice, there is usually a deviation between the two points. The distance between two points is the loss of feature points consistency. In this paper, we use SIFT [30] algorithm to extract feature points. SIFT algorithm has the characteristics of rotation invariance and scale invariance and is robust to illumination changes. The quality of SIFT feature point matching is relatively high. Although the speed is relatively slow, it is only used in model training and does not affect the speed of 3D reconstruction.

The quality of feature point detection and matching is directly related to the texture of the image. As shown in Figure 6, the two scenarios “Scan2” and “Scan92” are in the DTU dataset. The “Scan2” image is richly textured, while the “Scan92” image is relatively smooth. All scenes in DTU share 49 camera perspectives. In both scenes, we selected images from two camera perspectives for SIFT feature point detection and matching. The number of matched feature point pairs in Scan2 is much higher than that in scan92. Therefore, if the sum of position offsets of all feature points is taken as the error, the impact of the high-texture scenes on the network will be much greater than that of the low-texture scenes. This will reduce the generalization of the network. In addition, as shown in Figure 7, SIFT also produces wrong feature point matching for some images with repeated textures. If the average value of the position deviation of feature points is taken as the error, the accuracy will also be reduced. Therefore, we take the median of the position deviation of feature points as the feature points consistency error of two images.

When calculating the feature point consistency error, SIFT feature points are first detected and matched on the reference image (denoted as image 0) and the *i*-th source image. The matching point pair $\left(p_{0,j},\;p_{i,j}\right)$ is calculated. According to Equation (1), the backprojection point $p_{i,j}^{\prime }$ of $p_{0,j}$ in the *i*-th source image can be calculated. Then, the feature point consistency error of the *j*-th feature point is:
(7)$$error_{i,j}=∥p_{i,j}^{\prime }-p_{i,j}{∥}_{2}$$

The feature points consistency error between the reference image and the *i*-th source image is:
(8)$$error_{i}=median\left[error_{i,0},error_{i,1},\ldots ,error_{i,N_{s}}\right]$$
where $N_{S}$ represents the number of SIFT feature points between the reference image and the *i*-th source image.

Therefore, the feature points consistency loss function is defined as:
(9)$$L_{FC}=\overset{N}{\underset{i=2}{\sum }}error_{i}$$
where *N* represents the number of source images.

#### 2.2.4. High-Resolution Loss

At present, there are some multi-scale MVS networks similar to the one proposed in this paper [16,25]. All of them adopt the sum of losses at different scales as the total loss of the network. However, in the test process, the image resolution is relatively high. Therefore, in this paper, the depth images predicted in each stage are upsampled to high-resolution images, and then the loss of various measures is calculated.

The photometric consistency loss and semantic consistency loss are calculated at each stage. Therefore, the predicted depth map is firstly upsampled to the resolution of stage 0 by bilinear interpolation at each stage and then calculate the loss. The total loss is the sum of the losses at each stage.

The total photometric consistency loss is calculated as follows:
(10)$$L_{PC\_total}=\overset{3}{\underset{k=0}{\sum }}\overset{n_{k}}{\underset{i=1}{\sum }}L_{PC_{i}}^{k}$$
where *k* represents the number of stages and $n_{k}$ represents the number of iterations of stage *k*.

The total semantic consistency loss is calculated as follows:(11)$$L_{SC\_total}=\overset{3}{\underset{k=0}{\sum }}\overset{n_{k}}{\underset{i=1}{\sum }}L_{SC_{i}}^{k}$$
where *k* represents the number of stages and $n_{k}$ represents the number of iterations of stage *k*.

Feature point consistency loss is based on image feature point detection and matching. In the experiment, we find that the accuracy of feature point detection and matching decreases greatly when the image resolution is low. Therefore, the feature point consistency loss is calculated only at stage 0. The total feature point consistency loss is calculated as follows:
(12)$$L_{FC\_total}=L_{FC}$$

Finally, the total loss of the network is calculated as follows:
(13)$$L=\lambda_{1}L_{PC\_total}+\lambda_{2}L_{SC\_total}+\lambda_{3}L_{FC\_total}\;$$
where, $\lambda_{1}$, $\lambda_{2}$ and $\lambda_{3}$ are set to 0.8, 0.1, and 0.1 respectively.

## 3. Results

Firstly, we conducted a comparative experiment based on the DTU dataset [10] to comprehensively evaluate the performance of Unsup_patchmatchnet, including the 3D reconstruction effect, running memory usage, and time consumption. Secondly, the ablation experiment is conducted to analyze the influence of each module on the network performance. Finally, the generalization performance of the network is verified based on the Tanks and Temples dataset.

### 3.1. Performance Evaluation Based on DTU

The DTU dataset is an indoor multi-view stereo dataset containing 124 different scenes. All scenes share the same 49 camera views. Each view contains seven light variations. The segmentation method of the training set, test set, and validation set used in this paper are the same as that used in most previous MVS networks [13,17,18,19,24].

#### 3.1.1. Implementation Details

We designed the network using PyTorch and trained it using only the DTU training set. The parameter Settings of the depth estimation module are the same as those of PatchmatchNet. The number of source images is set to 4. During model training, the input image resolution is 640 × 512. The images are obtained from the center clipping of the original image after downsampling. The depth sampling range of the images is 425 mm to 935 mm. We trained in parallel on four Nvidia GTX 1080Ti GPUs using four batches for a total of 40 epochs. During the network test, the input image resolution is 1600 × 1200. After depth prediction, 3D point clouds of each scene are reconstructed for network performance evaluation.

#### 3.1.2. Result on DTU Dataset

The reconstructed 3D point cloud was evaluated according to the evaluation criteria provided by the DTU dataset. DTU evaluation criteria mainly include accuracy (Acc.), completeness (Comp.), and overall. Accuracy (Acc.) is measured by the distance between the reconstructed 3D point cloud and the real object point cloud, indicating the accuracy of the reconstructed points. completeness (Comp.) indicates the completeness of the reconstructed surface of the object; Overall is the average of accuracy and completeness and is a comprehensive standard of error. The smaller the value of the three criteria, the smaller the error of the reconstructed point cloud.

In this paper, we compare some recent traditional geometric methods, supervised and unsupervised learning MVS networks. The results are shown in Table 1. The overall performance of Unsup_patchmatchnet is still some distance from that of the best-supervised learning networks, but it exceeds the traditional geometric methods and other unsupervised learning networks. Compared with supervised learning networks, Unsup_patchmatchnet has a significant gap in both accuracy and completeness. For example, compared with PatchmatchNet, the accuracy error and completeness error are improved by 0.086 mm and 0.212 mm, respectively. Compared with traditional geometric methods, the accuracy error of Unsup_patchmatchnet is not the lowest. It is 0.23 mm higher than Gipuma and 0.171 mm higher than Tola. However, the completeness has obvious advantages. It is 0.384 mm lower than that of Gipuma and 0.701 mm lower than that of Tola. Therefore, the overall performance exceeds that of traditional geometric methods. Compared with other unsupervised learning networks, our network outperforms them in both accuracy and completeness. The completeness of the reconstructed model improved relatively little. Compared with MVS2, the completeness error is only reduced by 0.026 mm. However, the accuracy stands out. The accuracy error is more than 0.123 mm lower than that of other networks. Ultimately, Unsup_patchmatchnet outperforms several other MVS networks in terms of overall performance. Figure 8 shows some reconstructed 3D point cloud comparisons. It can be seen intuitively from the 3D point cloud that the details of the 3D point cloud reconstructed by Unsup_patchmatchnet are richer and more complete.

#### 3.1.3. Memory and Run-Time Comparison

In practical applications, MVS mainly runs the test process of the network, that is, the 3D reconstruction process. Therefore, we compare the memory usage and time consumption of Unsup_patchmatchnet with other networks in the testing process. Memory usage is the maximum memory requirements during testing, and time consumption is the average time to estimate depth once.

We compared some of the most advanced supervised MVS(MVSNet [14], CVP-MVSNet [16], PatchmatchNet [25]) and unsupervised MVS(M3VSNet [19], JDACS-MS [20]). The resolution of all network input images is set to 1600 × 1200. The result is shown in Figure 9. In terms of memory usage, MVSNet, CVP-MVSNet, M3VSNet, and JDACS-MS all have more memory usage due to 3DCNN regularization. PatchmatchNet and Unsup_patchmatchnet adopt PatchMatch mode, which significantly reduces memory usage and is only 1/5 of that of the other four networks. In terms of time consumption, 3DCNN regularization consumed more time. However, CVP-MVSNet and JDACS-MS also adopt the cascade mode based on 3DCNN regularization, so the time consumption is larger. The time consumption of PatchmatchNet and Unsup_patchmatchnet for depth prediction based on PatchMatch is only 1/2 of that of MVSNet and M3VSNet, and 1/6 of that of CVP-MVSNet and JDACS-MS. Unsup_patchmatchnet and PatchmatchNet have the same memory and time consumption.

### 3.2. Ablation Studies

#### 3.2.1. Effect of Different Loss Modules

This section mainly analyzes the influence of different loss function modules in Unsup_patchmatchnet on network performance. The effects of semantic consistency loss, feature point consistency loss, and high-resolution loss modules on network performance are compared. It is divided into 4 cases:

(a) None of the three modules (only the photometric consistency loss).

(b) Only semantic consistency loss.

(c) Semantic consistency loss and feature point consistency loss.

(d) All three types of modules.

Except for the above modules, other parameter Settings are the same as those in Section 3.1. Table 2 shows the evaluation and comparison of the reconstruction effect on the DTU test set under the four conditions. It can be seen that adding each module to the loss function improves the accuracy and completeness of the reconstruction effect to a certain extent. After adding the semantic consistency loss module, the overall error is reduced by 10.61%. This indicates that semantic consistency loss can further optimize the predicted image depth by constraining the semantic consistency between pixels of images from different perspectives. After adding the feature point consistency module, the overall error is reduced by 7.96%. This shows that the accuracy of the predicted depth map can be further improved by constraining the position offset of feature point pairs between images from different perspectives. After adding the high-resolution loss module, the overall error is reduced by 11.48%. The resolution of the image used in the test is higher than that used in the training. The high-resolution loss method enables the network trained with low-resolution images to reconstruct high-resolution images.

Figure 10 shows the comparison of reconstructed 3D point clouds in four cases. It can be seen intuitively from the figure that the point cloud is gradually complete and rich after the three modules are added in turn. This proves the effectiveness of semantic consistency loss, feature point consistency loss, and high-resolution loss modules.

#### 3.2.2. Effect of High-Resolution Loss

The effectiveness of the high-resolution loss method has been demonstrated. This section changes the resolution adjustment of depth maps at different scales to analyze the impact of different high-resolution loss calculation methods on network performance. As shown in Figure 11, we divided four cases for comparison:

The network performance comparison results of the four up-sampling methods of depth maps are shown in Table 3. Compared with no high-resolution loss (mode (a)), the overall reconstruction error of mode (b), mode (c) and mode (d) are reduced by 3.53%, 7.95% and 11.48%, respectively. Therefore, it can be concluded that the depth map resolution is positively correlated with the accuracy and completeness of the reconstructed 3D point cloud. The high-resolution loss method is proven to be effective.

### 3.3. Generalization Ability on Tanks and Temples

In this section, the Tanks and Temples dataset is used to test the generalization of the network. We use model parameters trained on the DTU dataset. No adjustments were made to the model. The input image size is set to 1920 × 1056. The number of source images is set to 6. The test results are shown in Table 4. The data in the table are F scores, and the higher the score, the better the network performance.

As can be seen from Table 4, in most scenes, the reconstruction effect of Unsup_patchmatchnet is better than that of MVS2 and M3VSNet. In particular, the scores of “Francis”and “Train” scenarios were twice and 1.5 times higher than those of the other two network models, respectively. Although the scores in the “M60” and “Panther” scenarios are not the highest, the difference is not large, which is 11.34% and 8.37% lower than the highest scores in the other two networks, respectively. Finally, the overall average score of Unsup_patchmatchnet exceeds the other two network models. Figure 12 shows the reconstructed point cloud. It can be seen that no matter whether considering smaller objects (such as “Family” and “Horse”) or larger scenes (such as “Playground” and “Lighthouse”), Unsup_patchmatchnet can be reconstructed well. For some non-ideal areas, such as the sand in “Playground”, it can also be well presented. In conclusion, it can be proved that Unsup_patchmatchnet has strong generalization properties.

## 4. Discussion

The results in Table 1 fully prove the effectiveness of our proposed network. It goes beyond most current methods of unsupervised learning. The improvement of accuracy is obvious, but the improvement of integrity is relatively small. However, compared with supervised methods, the gaps in accuracy and completeness are obvious. This reflects the unsupervised learning approach still faces great challenges. In addition, although current deep learning methods are more comprehensive than traditional geometric methods, the accuracy of traditional geometric methods is very high. Our proposed network incorporates feature point consistency loss. This is essentially a combination of deep learning methods and traditional methods. This is the reason why the accuracy of Unsup_patchmatchnet improved significantly compared with other unsupervised learning methods.

We demonstrate that our method has the advantages of fast running time and low memory usage in depth estimation. That is, we are comparing the performance of the network in the test. Because the actual use is mainly in the process of testing, the model will not be retrained. However, our method takes a long time to train. All unsupervised learning methods take more time to calculate the loss function. The feature point consistency loss proposed by us requires feature point detection and matching of the image, so it consumes more time. We extract and save the feature point information of the image in advance, thus greatly reducing the training time.

The results in Table 3 demonstrate the effectiveness of the high-resolution loss method. When the DTU data set is used for model training, the resolution of the input image is generally 640 × 512. However, at the time of the test, the input image resolution was 1600 × 1200. The resolution of the training image is much lower than that of the test image. We haven’t changed the resolution of the input image. Because this is more conducive to comparison with other methods, but also limited by memory. We upsample the predicted depth map. This is equivalent to improving the resolution of the training image. However, the depth map resolution is also limited by memory.

We analyzed the reasons for the poor reconstruction quality of “M60” and “Panther” scenes. The network designed in this paper adopts the feature point consistency loss function. This makes the depth predicted by the pixel at the feature point more accurate. The network proposed in this paper estimates image depth based on PatchMatch. It propagates the depth of the feature points to the surrounding pixels. Due to the complex textures in the “M60” and “Panther” scenes, the feature points are dense. This results in some smooth areas with more alternative depths. Therefore, the 3D points calculated from the same position of objects in different viewing angles have large errors. Points with large errors will be filtered in the reconstruction. The reconstructed point cloud tends to produce more cavities. The integrity of the reconstructed point cloud will drop.

The proposed feature point consistency loss also has some limitations. As can be seen from Figure 5, some low-texture images have few or no feature points. Therefore, in some low-texture scenes, feature point consistency loss is not effective.

## 5. Conclusions

Based on the idea of PatchMatch, in this paper, we propose an unsupervised learning multi-view 3D reconstruction network Unsup_patchmatchnet. The network incorporates multi-metric high-resolution joint loss such as photometric consistency, semantic consistency, and feature point consistency. Experiments on DTU datasets show that Unsup_patchmatchnet is superior to some current unsupervised multi-view 3D reconstruction networks in accuracy and completeness. Unsup_patchmatchnet has the advantages of less memory usage and less time consumption. The test of different datasets also verifies that Unsup_patchmatchnet has good generalization properties. In the future, we will focus on 3D reconstruction of low texture objects. This is a difficult problem in both traditional and deep learning methods. However, in real life, low-texture scenes are very common. Therefore, line features and plane features can be used to replace point features in image feature extraction. This way of using the overall feature can use more image information, can better adapt to the situation of low texture. In addition, although the network proposed in this paper has fast computing speed and low memory consumption, it is still unable to achieve real-time 3D reconstruction on the mobile terminal. Therefore, we will continue to lightweight the network structure in the future to improve its utility.

## Figures and Tables

**Figure 1 sensors-23-00136-f001:**
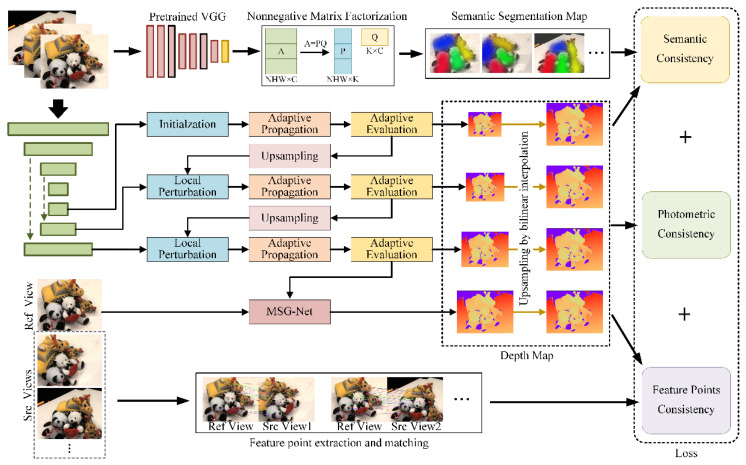
The network structure of Unsup_patchmatchnet.

**Figure 2 sensors-23-00136-f002:**
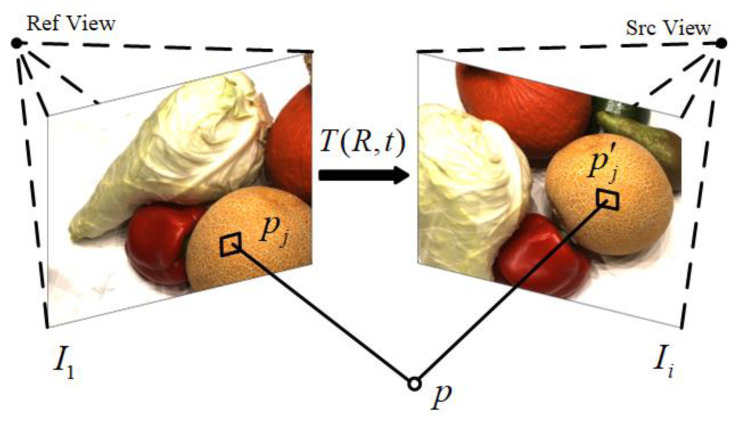
Principle of photometric consistency loss.

**Figure 3 sensors-23-00136-f003:**
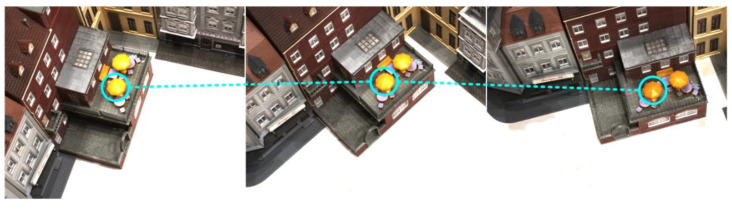
The photometric difference of different visual angles under the same illumination.

**Figure 4 sensors-23-00136-f004:**
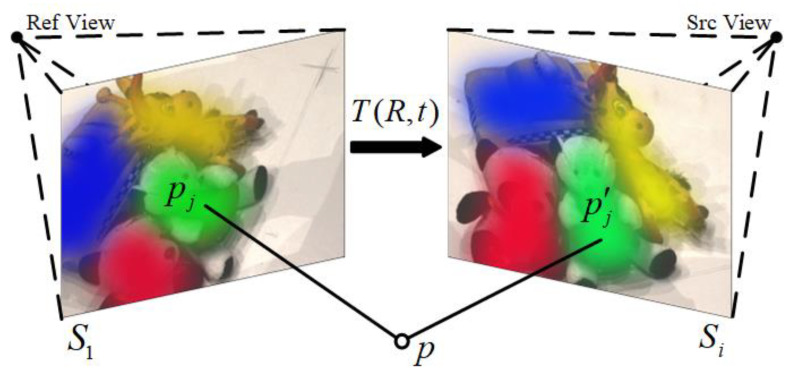
Principle of semantic consistency loss.

**Figure 5 sensors-23-00136-f005:**
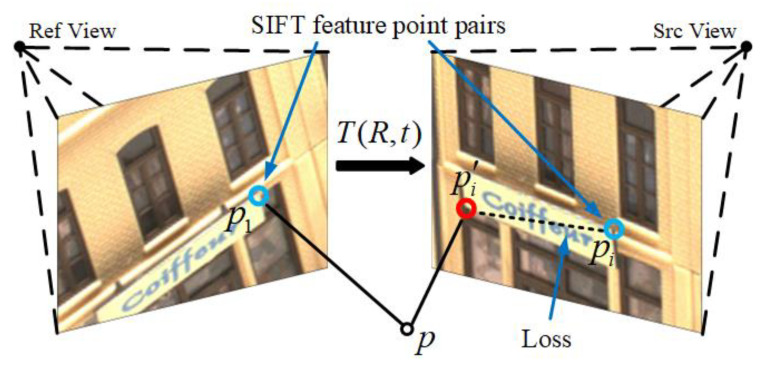
Principle of feature point consistency loss.

**Figure 6 sensors-23-00136-f006:**
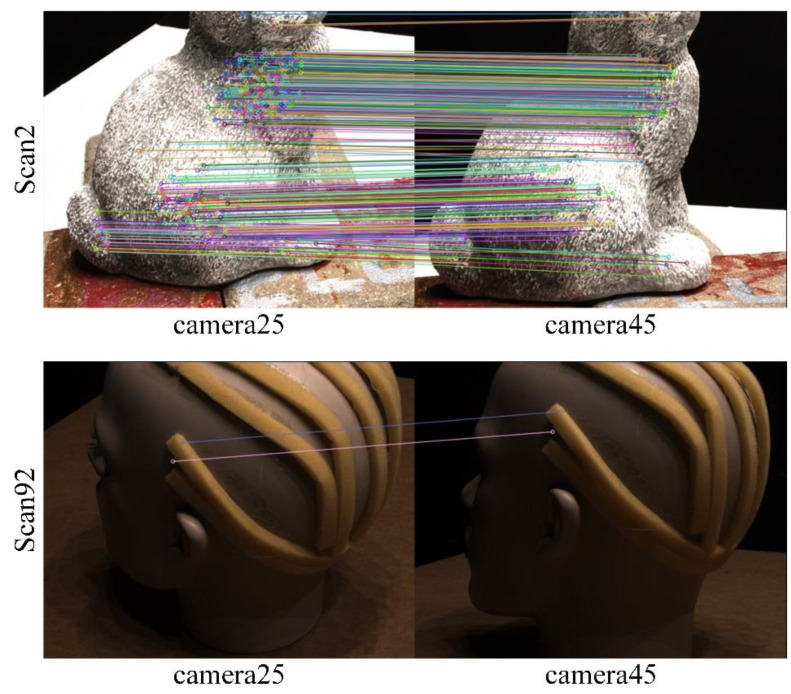
Effect of image texture on SIFT feature point extraction and matching.

**Figure 7 sensors-23-00136-f007:**
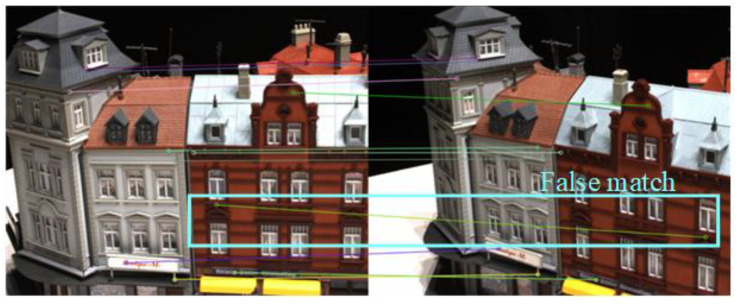
Incorrect feature point matching in images with repeated textures.

**Figure 8 sensors-23-00136-f008:**
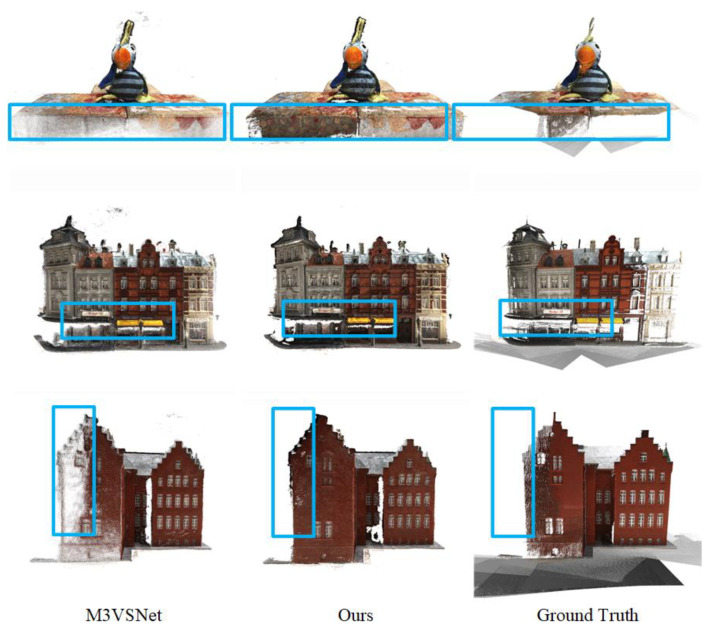
Comparison of reconstructed 3D point clouds.

**Figure 9 sensors-23-00136-f009:**
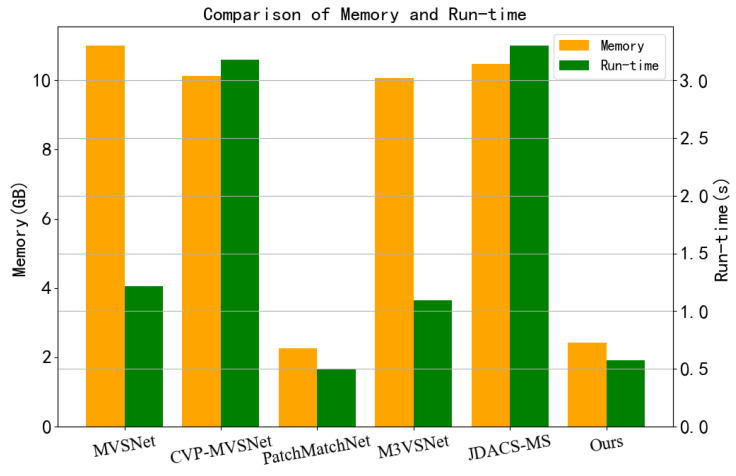
Comparison of memory usage and time consumption.

**Figure 10 sensors-23-00136-f010:**
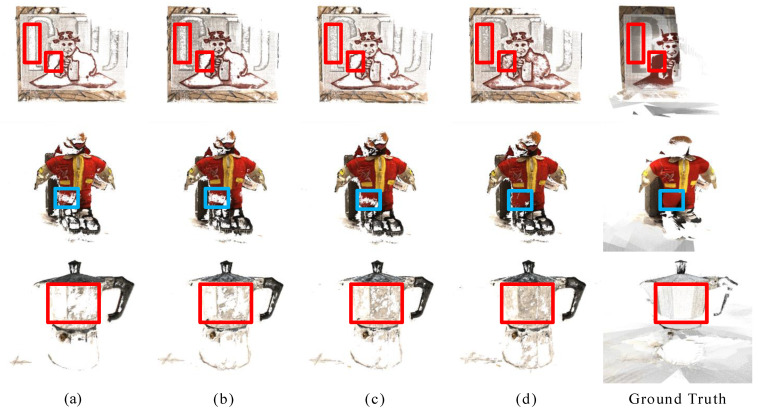
Comparison of reconstruction results after adding each module. (**a**–**d**) has the same meaning as in Table 2.

**Figure 11 sensors-23-00136-f011:**
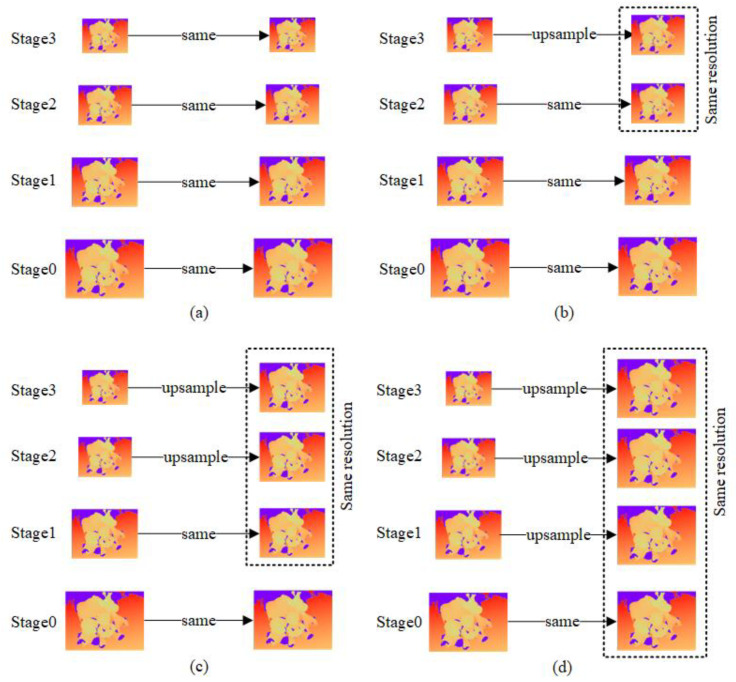
Different depth map adjustment methods. (**a**) The resolution of the depth map at each stage remained unchanged; (**b**) The resolution of the depth map at stage 3 is upsampled to the size of the depth map at stage 2. The resolution of the depth map from stage 0 to stage 2 remains the same; (**c**) The resolution of the depth maps at stage 2 and Stage 3 is upsampled to the size of the depth map at stage 1. The resolution of the depth map at stage 0 and stage 1 remains the same; (**d**) The resolution of the depth map from stages 1 to 3 is upsampled to the size of the depth map at stage 0. The resolution of the depth map at stage 0 remains the same.

**Figure 12 sensors-23-00136-f012:**
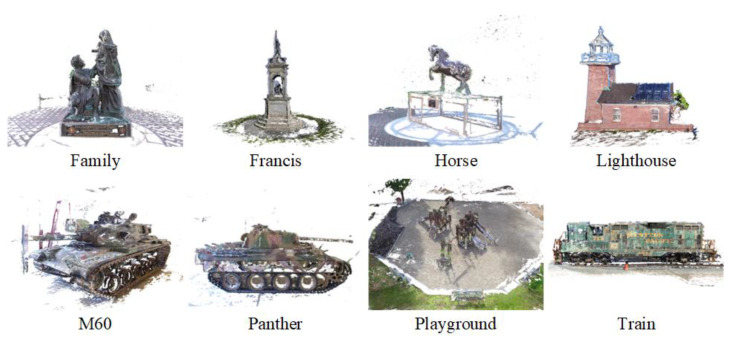
Reconstruction result on Tanks and Temples.

**Table 1 sensors-23-00136-t001:** Quantitative results on DTU evaluation benchmark. Geo. represents traditional geometric methods. Sup. represents supervised methods. UnSup. represents unsupervised methods.

Method		Acc. (mm)	Comp. (mm)	Overall (mm)
Geo.	Furu [6]	0.613	0.941	0.777
	Tola [31]	0.342	1.190	0.766
	Camp [32]	0.835	0.554	0.694
	Gipuma [8]	0.283	0.873	0.578
Sup.	Surfacenet [12]	0.450	1.040	0.745
	MVSNet [14]	0.444	0.741	0.592
	R-MVSNet [21]	0.383	0.452	0.417
	PatchmatchNet [25]	0.427	0.277	0.352
UnSup.	Unsup_MVS [17]	0.881	1.073	0.977
	MVS^2^ [18]	0.760	0.515	0.637
	M^3^VSNet [19]	0.636	0.531	0.583
	Ours	**0.513**	**0.489**	**0.501**

**Table 2 sensors-23-00136-t002:** Comparison of effect of each module.

Method	Semantic Consistency Loss	Feature Point Consistency Loss	High-Resolution Loss	Acc. (mm)	Comp. (mm)	Overall (mm)
(a)	×	×	×	0.714	0.662	0.688
(b)	√	×	×	0.696	0.534	0.615
(c)	√	√	×	0.613	0.519	0.566
(d)	√	√	√	**0.513**	**0.489**	**0.501**

√ indicates that the module is included, × indicates that the module is not included.

**Table 3 sensors-23-00136-t003:** Effect of different depth map adjustment methods.

Method	Acc. (mm)	Comp. (mm)	Overall (mm)
(a)	0.613	0.519	0.566
(b)	0.581	0.511	0.546
(c)	0.537	0.505	0.521
(d)	**0.513**	**0.489**	**0.501**

**Table 4 sensors-23-00136-t004:** Result on Tanks and Temples.

Method	Mean	Family	Francis	Horse	Lighthouse	M60	Panther	Playground	Train
MVS^2^	37.27	47.74	21.55	19.50	44.54	**44.86**	**46.32**	43.38	29.72
M^3^VSNet	37.67	47.74	24.38	18.74	44.42	43.45	44.95	47.39	30.31
Ours	**43.91**	**49.00**	**48.56**	**25.74**	**49.52**	39.77	42.44	**51.14**	**45.15**

## Data Availability

Publicly available datasets were analyzed in this study. These data can be found here: http://roboimagedata.compute.dtu.dk/?page_id=36 (DTU, accessed on 15 July 2022), https://www.tanksandtemples.org/download/ (Tanks and temples, accessed on 22 July 2022).

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
