# Peer review of "Unsupervised 3D Reconstruction with Multi-Measure and High-Resolution Loss"

_sensors, 2022, doi:10.3390/s23010136_

Round 1

Reviewer 1 Report

In this manuscript, an unsupervised 3D reconstruction with multi-measure and high-resolution loss has been proposed. It is the interesting topic; however, there are fundamental problems that authors should consider as follow:

1.       The abstract should be rewritten. Especially, the numerical results could be added.

2.       In the introduction, the innovation of the manuscript is well discussed, but it is better to add the related work section to the introduction and also use the following reference to enrich the work:

·        10.1109/JSTARS.2022.3189528

3.       The tables should be analyzed more qualitatively in the discussion section.

4.       What have been the limitations of your work?

5.       The conclusion is very brief. Suggestions for future works should be added in this section.

Reviewer 2 Report

This paper proposed an end-to-end unsupervised multi-view 3D reconstruction network framework based on PatchMatch, Unsup_patchmatchnet. It is of certain interest to the journal readers. However, in the experimental section, the results are not very convincing. More experiments with similar methods should be conducted with the results fully explained. 

Reviewer 3 Report

In this article the authors present an interesting contribution in the field of unsupervised 3D reconstruction. Overall, the article is well written and structured, although there are some issues that must be improved:

1. The sections of the article do not conform to what is indicated in the template. For example, section 2 (Related work) should be merged with section 1 (Introduction). Section 2 should be 'Materials and methods' and section 3 'Results. In general, authors should review this aspect.

2. The authors should explain in some more detail the advantages of unsupervised methods, such as the one they propose, over supervised methods.

3. The results shown in Table 4 show that the method proposed by the authors provides the best results, except in two of the cases (M60 and Panther). Do the authors have any explanation as to why in these two cases their method is the one with the worst results?

Round 2

Reviewer 2 Report

The related issues have been well addressed and this paper can be published as it is.